# Review of Disease-Specific microRNAs by Strategically Bridging Genetics and Epigenetics in Oral Squamous Cell Carcinoma

**DOI:** 10.3390/genes14081578

**Published:** 2023-08-02

**Authors:** Iphigenia Gintoni, Stavros Vassiliou, George P. Chrousos, Christos Yapijakis

**Affiliations:** 1Unit of Orofacial Genetics, 1st Department of Pediatrics, National Kapodistrian University of Athens, “Aghia Sophia” Children’s Hospital, 115 27 Athens, Greece; iph.gintoni@gmail.com; 2Department of Molecular Genetics, Cephalogenetics Center, 176 72 Athens, Greece; 3Department of Oral and Maxillofacial Surgery, School of Medicine, National Kapodistrian University of Athens, Attikon Hospital, 124 62 Athens, Greece; stvasil@med.uoa.gr; 4University Research Institute for the Study of Genetic and Malignant Disorders in Childhood, Choremion Laboratory, “Aghia Sophia” Children’s Hospital, 115 27 Athens, Greece; chrousge@med.uoa.gr

**Keywords:** OSCC, oral cancer, genes, mutations, miRNA, expression, mir-34a, mir-155, mir-124, mir-1, mir-16, bioinformatics, in silico analysis

## Abstract

Oral squamous cell carcinoma (OSCC) is one of the most prevalent human malignancies and a global health concern with a poor prognosis despite some therapeutic advances, highlighting the need for a better understanding of its molecular etiology. The genomic landscape of OSCC is well-established and recent research has focused on miRNAs, which regulate gene expression and may be useful non-invasive biomarkers or therapeutic targets. A plethora of findings regarding miRNA expression have been generated, posing challenges for the interpretation and identification of disease-specific molecules. Hence, we opted to identify the most important regulatory miRNAs by bridging genetics and epigenetics, focusing on the key genes implicated in OSCC development. Based on published reports, we have developed custom panels of fifteen major oncogenes and five major tumor suppressor genes. Following a miRNA/target gene interaction analysis and a comprehensive study of the literature, we selected the miRNA molecules which target the majority of these panels that have been reported to be downregulated or upregulated in OSCC, respectively. As a result, miR-34a-5p, miR-155-5p, miR-124-3p, miR-1-3p, and miR-16-5p appeared to be the most OSCC-specific. Their expression patterns, verified targets, and the signaling pathways affected by their dysregulation in OSCC are thoroughly discussed.

## 1. Introduction

The oral cavity is the most prevalent site for malignancies of the gastrointestinal and upper respiratory tracts [1]. Characterized by the uncontrolled growth of abnormal cells in the oral cavity, including the lips, tongue, gums, and lining of the cheeks, oral cancer is one of the most common human malignancies, ranking sixth in prevalence worldwide with an estimated global incidence of more than 377,700 new cases in 2020 [2]. More than 95% of diagnosed oral cancer cases are represented by oral squamous cell carcinoma (OSCC), which arises from the stratified squamous epithelial layer of the oral mucosa [3,4]. OSCC is a significant global health concern, with alarming mortality rates of more than 60% [5], largely due to the fact that over 50% of patients are diagnosed in advanced stages (III and IV) and exhibit lymph node infiltration [5,6,7]. Consequently, despite advances in therapeutic approaches such as chemotherapy, radiation, and surgical excision, OSCC mortality rates have remained exceptionally high for a minimum of two decades [8]. Recurrence rates of OSCC are also high, with up to 45% of patients relapsing and facing survival odds of less than 10% [9,10], highlighting the need for further understanding of OSCC as a complex and heterogeneous disease involving the well-established dysregulation of multiple genes and several, currently explored, epigenetic signatures [11,12]. In accordance with other malignancies, OSCC is characterized by the pathological dysregulation of the cell cycle [13]. Although its genetic landscape can be diverse, several genes stand out, demonstrating unique mutational patterns, while others are reported to exhibit characteristically abnormal expression levels within the tumor’s microenvironment [14].

Beyond genetics, the role of epigenetic alterations to OSCC has been under investigation in recent years, with a particular focus on microRNAs (miRNAs), a class of small non-coding RNAs that have emerged as essential regulators in OSCC development and progression. MiRNAs regulate post-transcriptional gene expression by binding to the 3’ untranslated regions (UTRs) of target messenger RNAs (mRNAs), causing their degradation or translational suppression [15]. MiRNAs are significantly overexpressed or downregulated in malignant tissues compared to normal tissues, presenting as tumor-suppressing or oncogenic or epigenetic factors (oncomiRs), depending on whether they inhibit oncogene or tumor-suppressor gene expression [16].

MiRNA dysregulation has been identified in OSCC, leading to abnormal gene expression patterns that are associated with oral carcinogenesis. Numerous studies have revealed multiple miRNAs that are involved in essential biological processes such as cell proliferation, apoptosis, invasion, metastasis, and angiogenesis [17]. Although genetic and epigenetic mechanisms were initially thought to be discrete, it is now known that they present a strong interdependent relationship, which, if decoded, could fill the gaps and assist in mapping the overall molecular signatures of OSCC, resulting in a better understanding of its etiology, decidedly reliable biomarkers or even targeted epigenetic therapeutics [12].

The current research on OSCC-related miRNA expression has resulted in a tremendous pool of data implicating hundreds of significantly dysregulated distinct miRNAs [17,18]. Therefore, we aimed to reveal through strategic bioinformatic filtering those epigenetically important molecules that stand out by targeting and influencing the expression of key genes known to govern oral carcinogenesis. We review here the implication of the five most important miRNA molecules in OSCC by discussing their expressional dysregulation in OSCC tissue, cell lines, and patients’ biofluids (e.g., saliva, whole blood, serum, and plasma) and their influence on the post-transcriptional expression of their verified target genes, as well as on the signal transduction pathways that are subsequently affected by their dysregulation.

## 2. Methodology

We report our evaluation of the existing literature about OSCC development through bioinformatic analysis that combines the most significant mutant oncogenes and tumor suppressor genes with the most implicated miRNAs whose expression is dysregulated in OSCC. Initial identification of multiple tens of reportedly involved genes was conducted, followed by subsequent analyses in order to highlight the molecules of extreme importance for oral carcinogenesis that might ultimately be utilized as prognostic and diagnostic markers for that cancer type.

### 2.1. Selection of Most Significant Implicated Genes in OSCC

An assortment of twenty key cell-cycle regulatory genes was selected through bioinformatic analysis on miRNA/target prediction, including the tumor suppressor genes *TP53*, *CDKN2A*, *FAT1*, *CASP8*, *PTEN*, and the oncogenes *NOTCH1*, *HRAS*, *PIK3CA*, *EGFR*, *ERBB2*, *FGFR1*, *FGFR2*, *FGFR3*, *FGFR4*, *FGF2*, *ETS1*, *JUN*, *MKI67*, *MYC,* and *BCL2*, which are strongly and typically associated with OSCC development in humans and laboratory animals [14,19,20,21,22,23], as discussed in detail below.

### 2.2. Selection of Most Significant miRNAs in OSCC

We conducted a PubMed search using the following keyword combinations: (“miRNAs” AND “OSCC”), (“miRNAs” AND “expression” AND OSCC”), and (“miRNAs” AND “expression” AND OSCC”), limiting our search to review articles in English, published between 1 January 2020 and 20 June 2023. These searches yielded 39, 23, and 63 results, respectively, from which we selected 13 articles that were specifically focused on OSCC (excluding precancerous oral pathologies), included more than three implicated miRNA molecules, and clearly stated the expression patterns, as well as the sample sources that corresponded to miRNA quantification results (Table 1 and Table 2). After compiling miRNA-expression data from each review, we identified 106 miRNA molecules that are significantly upregulated in OSCC-related biological samples (tissue, cell lines, saliva, whole blood, serum, or plasma) (Table 1) and 133 that are considerably downregulated (Table 2). Those results provided a detailed picture of the current literature on miRNA expression patterns in OSCC and served as a data pool for the following step of the process, which comprised bioinformatic filtering.

### 2.3. Analysis of Target Gene/miRNA Interaction

For target identification and miRNA/target interaction analysis, we developed two miRNA regulatory networks using the “miRNet 2.0” miRNA-centric network visual analytics software (https://www.mirnet.ca/miRNet/home.xhtml, accessed on 5 July 2023, Xia Lab/McGill University, Montréal, QC, Canada). One network identified all miRNA molecules that are predicted to target and, therefore, may regulate the expression of the 5 selected tumor suppressor OSCC-driver genes *TP53*, *CDKN2A*, *FAT1*, *CASP8,* and *PTEN* (Figure 1). The second network comprised all the miRNA molecules that target and might affect the expression of the 15 selected driver oncogenes *NOTCH1*, *HRAS*, *PIK3CA*, *EGFR*, *ERBB2*, *FGFR1*, *FGFR2*, *FGFR3*, *FGFR4*, *FGF2*, *ETS1*, *JUN*, *MKI67*, *MYC* and *BCL2* (Figure 2). Afterward, following the examination of all the OSCC-related miRNAs listed in Table 1 and Table 2, we selected from the networks 2 miRNA molecules that target more than 3 (60%) of the 5 tumor suppressor genes and are reported to be significantly upregulated in OSCC, and 4 miRNAs that target at least 9 of the 15 (60%) selected oncogenes and at the same time exhibit significant downregulation in OSCC. One miRNA was included in both groups of OSCC-driver genes. The 5 miRNAs obtained from the aforementioned filtering were analyzed as to their expression patterns in OSCC-related biological materials (e.g., tumor tissue, cell lines, saliva, whole blood, serum, or plasma samples), their target genes of interest and the possible influence of their dysregulation in signaling pathways that are known to be involved in tumorigenesis and/or OSCC development specifically.

### 2.4. Important Driver Genes in OSCC

Twenty genes were selected to serve as an OSCC-specific signature panel in this study due to their significant involvement in key biological processes and pathways relevant to oral oncogenesis, which is supported by extensive research on human patients and laboratory animal models [14,19,20,21,22,23,33]. The dysregulation of these genes contributes to OSCC pathogenesis because their protein products play crucial roles in cell cycle regulation and various other cell functions.

### 2.5. Tumor Suppressor Genes

*TP53*, also known as the p53 gene, encodes the protein p53, which acts as a tumor suppressor since it plays a crucial role in regulating cell cycle progression, DNA repair, apoptosis, and genomic stability [34,35]. *TP53* is considered to be the most mutated gene in OSCC, resulting in uncontrolled cell growth and genomic instability [36]. Inactivation of *TP53* has been associated with aggressive OSCC traits, advanced tumor stage, resistance to radiation and chemotherapy, increased risk of recurrence, and lower overall survival [19,36,37,38,39].

The *CDKN2A* gene encodes cyclin-dependent kinase inhibitor 2A, also referred to as p16INK4a or p16. CDKN2A prevents the progression of the cell cycle by inhibiting the activity of cyclin-dependent kinases [40]. *CDKN2A* is an additional tumor suppressor gene that is frequently mutated in OSCC, promoting uncontrolled cell proliferation [14,37], particularly in non-smoking patients [36]. It is believed to promote malignant transformation and disease progression [41], which is supported by the observation that infection with high-risk HPV strains induces overexpression of p16 in oral premalignant lesions and carcinomas [14].

*FAT1* is a tumor suppressor gene that encodes the FAT atypical cadherin 1 protein, which is essential for cell–cell adhesion as well as tissue development and proliferation [42]. By modulating cell proliferation, apoptosis, and migration, FAT1 typically inhibits tumorigenesis. *FAT1* is frequently mutated or silenced in OSCC and is therefore regarded as one of the disease’s driver genes [42]. The loss of *FAT1* function has been associated with OSCC development, progression, and, ultimately, metastasis since it disrupts cell adhesion and promotes epithelial-to-mesenchymal transition (EMT) [43].

*CASP8* is a tumor suppressor gene that encodes the apoptosis-regulating enzyme caspase-8 [44]. Driven by cell surface receptors, the activation of caspase-8 leads to the elimination of abnormal cells, thereby maintaining tissue homeostasis and preventing cancer development. Loss-of-function mutations in *CASP8* have been associated with the development of OSCC and contribute to cancer-cell proliferation, survival, migration, and chemoresistance [44,45,46].

*PTEN* is a tumor suppressor gene that encodes the phosphatase and tensin homolog protein [47]. It maintains the delicate balance between cell growth, proliferation, and apoptosis by inhibiting the PI3K/AKT signaling pathway and thus regulates cell survival, proliferation, and apoptosis [13,47]. *PTEN* loss-of-function mutations are frequently observed in OSCC and contribute to its pathogenesis by stimulating cell proliferation and enhancing cell survival as a result of PIK3/AKT cascade subsequent upregulation [13,21]. In fact, loss or down-regulation of *PTEN* expression is typical for OSCC and is associated with a substantially poor prognosis for OSCC patients [13,47].

### 2.6. Oncogenes

*NOTCH1* codes for the transmembrane receptor NOTCH1 protein, which is involved in cell signaling pathways that govern cell fate determination, differentiation, and proliferation [48]. *NOTCH1* mutations are identified in 60% of OSCCs [14], and dysregulated NOTCH1 signaling can contribute to uncontrolled cell proliferation and OSCC initiation. OSCC tumors exhibit elevated NOTCH1 expression levels, which have been associated with OSCC progression and lymph node metastasis, whereas inhibition of the NOTCH pathway reduces cell proliferation and invasion. Moreover, the inactivation of the *NOTCH1* gene has been correlated with favorable disease outcomes [13,48,49].

*HRAS* is a proto-oncogene that encodes the H-ras small GTPase protein, which regulates the cell cycle in response to growth factor stimuli by activating a signaling cascade that leads to cell growth and division [50,51]. *HRAS* is considered to be one of the most frequently mutated genes in OSSC [13,19]. *HRAS* activating mutations may result in constitutive activation of downstream signaling pathways, enhancing cell proliferation and contributing to OSCC formation, invasion, and metastasis [50,51].

The *PIK3CA* gene encodes the p110 catalytic subunit of phosphatidylinositol 3-kinase (PI3K), a signaling protein involved in cell growth and survival via the PI3K/AKT signaling pathway. Gain-of-function mutations in *PIK3CA* induce constitutive activation of the PI3K/AKT signaling pathway, which promotes cell survival and proliferation and is associated with OSCC development and progression [52,53]. Furthermore, PI3K/AKT activation can interact with other oncogenic signaling pathways, such as RAS/MAPK and the EGFR, boosting OSCC progression [50,53,54]. Finally, PIK3CA expression has been found to be elevated in more than 50% of OSCC tumors, and *PIK3CA* gene copy amplification has been proposed as a potential prognostic marker, as it has been associated with poor patient survival [21].

The *EGFR* gene encodes the EGFR (epidermal growth factor receptor) protein, which is a receptor tyrosine kinase that regulates cell proliferation, survival, and differentiation by delivering extracellular growth factor signals. In OSCC, *EGFR* is considered an oncogene. Indeed, *EGFR* overexpression and activating mutations in its sequence result in abnormal EGFR-signaling, which promotes enhanced cell proliferation and survival, angiogenesis, and tumor growth while imposing resistance to EGFR-targeting therapies in OSCC [50,55]. Furthermore, *EGFR* gene mutations have been detected in 15% of HPV-negative and 8% of HPV-positive SCC of the head and neck, and EGFR elevated expression has been associated with a poor prognosis and a significantly decreased cancer-free survival in precancerous lesions [14].

*ERBB2*, also known as *HER2*, is an oncogene that encodes the erb-b2 receptor, a tyrosine kinase that impacts cell proliferation and differentiation. Cell cycle dysregulation, carcinogenesis, and tumor growth can result from *ERBB2* amplification or overexpression [56]. Overexpression of ERBB2 has been linked to increased proliferation, growth, invasion, and metastasis in OSCC, whereas patients with *ERBB2* mutations have been linked to significantly lower survival rates [50,57], and *ERBB2* copy-number amplification in OSCC cancer-free margins has been suggested to serve as a predictor of unfavorable prognosis [58].

The human fibroblast growth factor receptor (FGFR) family of the FGF superfamily is comprised of the *FGFR1*, *FGFR2*, *FGFR3,* and *FGFR4* genes, which encode the fibroblast growth factor receptor proteins −1, −2, −3, and −4, respectively. Extracellular signaling activates FGFRs on the cell membrane, resulting in proliferation, cell migration, and survival. FGFRs are considered to be oncogenic and have considerably increased expression in a variety of cancers, including OSCC [59,60]. FGFR signaling deviations due to gene amplification or activating mutations have been associated with OSCC development and progression [50,61]. FGFR3 and FGFR4 overexpression is seen in 48% and 41% of oral malignancies, respectively, while *FGFR1* mutations are identified in 10% of HPV-negative cases [14]. Finally, strong expression of FGFR3, as well as FGFR2 and its ligand FGF2 (fibroblast growth factor 2) in oral premalignant lesions, has been shown to be a positive predictor of malignant transformation [14,62]. Indeed, dysregulated FGF2 expression is prevalent in the early stages of oral carcinogenesis and is thought to exert fibrotic and angiogenic effects [63].

*ETS1* is a gene that encodes the Ets-1 protein, which belongs to the E26 transformation-specific (ETS) transcription factor family. *ETS1* regulates an array of biological processes, including cell proliferation, differentiation, and death. ETS1 has been shown to control signaling pathways involved in cell cycle progression and extracellular matrix remodeling, including the TGF-b pathway [64,65]. *ETS1* overexpression has been observed in OSCC tissues and is believed to promote OSCC development, EMT, lymph node infiltration, and distant metastasis, and has been associated with advanced OSCC stages and notably poor survival of patients [64,65,66].

The *JUN* proto-oncogene encodes c-Jun, a subunit of the AP-1 transcription factor that is involved in cellular processes such as proliferation, apoptosis, and differentiation and may influence the expression of cell cycle regulatory genes [39]. C-Jun is important in the initiation of OSCC, and high expression of c-Jun is associated with OSCC progression, invasion, and overall poor patient prognosis [39], as well as more aggressive tumor behavior in HPV-negative OSCCs [67]. In fact, the *JUN* gene’s expression is 2.69-fold higher in metastatic OSCCs compared to non-metastatic tumors (*p* = 0.012) and is associated with substantially lower survival rates [39], with high c-Jun expression considered predictive of OSCC-induced mortality [68]. Finally, high EGFR expression, which has been linked to OSCC development, activates c-Jun via the phospholipase Cy (EGFR-PLCy-Raf-MEK-ERK) signaling pathway, resulting in elevated c-Jun expression that gradually increases during OSCC progression in representative animal models [22].

The *MYC* gene, also known as the c-myc transcription factor, is involved not only in cell-cycle regulation through mediating cell proliferation, differentiation, and apoptosis but also in cell metabolism and tumor stem cell self-renovation. MYC functions as an oncogene in a range of cancer types, including OSCC, exhibiting activating mutations and aberrant expression [69]. The activation and upregulation of c-myc expression in response to TGF-b1 signaling in growth factors results in uncontrollable cancer-cell proliferation in OSCC and contributes to its progression [70,71]. Increased c-myc levels have been associated with advance-staged malignancies as well as an increased risk of lymph node infiltration and metastasis [69].

The *MKI67* gene encodes the Ki67 nuclear protein, which is not expressed in non-dividing cells and is thus characterized as a marker of proliferation levels in tumor tissues due to its notably high expression in cancer cells, which is associated with advanced tumor stages, disease progression, metastasis, and prognosis of several human neoplasms, including OSCC [72]. Ki67 expression increases alongside the degree of dysplasia in the oral mucosa, and ki67 positivity is found in all OSCC tumors, with significant expression intensity at invasion sites. Ki67 overexpression has been associated with high levels of differentiation in OSCC and is thought to be a valuable prognostic marker of tumor aggressiveness [73].

The *BCL2* gene encodes the Bcl-2 (B-cell lymphoma 2) protein, which regulates cell death by inhibiting the activation of apoptotic pathways and thereby promoting cell survival. The anti-apoptotic Bcl-2 is widely overexpressed in malignancies, including OSCC [74]. Enhanced Bcl-2 expression in tumor tissue has been observed in OSCC, leading to defective apoptosis and increased cell survival. Histological Bcl-2 overexpression in OSCC has been associated with advanced-stage carcinomas, lymph node metastases, and an unfavorable overall prognosis [74,75,76]. Bcl-2 has also been suggested as a good predictive indicator for malignant transformation in the initial stages of oral carcinogenesis, from normal oral mucosa to dysplastic lesions [75,77], while it is supported that its levels decline in well and moderately differentiated tumors in contrast with poorly differentiated OSCCs that exhibit bold Bcl-2 expression [77].

### 2.7. Combinatory Significance of Key Driver Genes in OSCC

Each of the aforementioned 20 genes, when in dysregulation, significantly contributes to OSCC pathogenesis by promoting biological processes such as cell proliferation, inhibition of apoptosis, cell survival enhancement, and activation of signaling pathways involved in tumor growth, invasion, and metastasis. However, comprehensive studies have revealed the combinatory role of large subsets of those genes and their protein products in oral oncogenesis in both humans and representative laboratory animals in terms of mutational and expressional profiles.

In fact, around 88% of OSCC patients harbor at least one mutation in *TP53*, *CDKN2A*, *FAT1*, *CASP8*, *NOTCH1*, *HRAS,* or *PIK3CA* genes in the primary OSCC tumor tissue. The same mutational profile is also detected in cell-free DNA extracted from saliva samples in 93.4% of the corresponding patients. This set of genes is suggested as a very promising gene panel for screening high-risk individuals for early detection, monitoring during treatment, as well as post-operative surveillance [20]. Driver mutations of the *PTEN* and *EGFR* genes are also reported as typical in OSCC development alongside mutations in the sequences aforementioned [19].

Additionally, a hamster model of sequential oral oncogenesis showed the increased expression of the transmembrane receptors EGFR, ERBB2, FGFR2, and FGFR3, as well as the increased expression of TP53, MYC, ETS1, MKI67, and JUN proteins in early stages of oral cancer indicating that these molecules may be used as early prognostic factors for the progression of OSCC [33]. On the other hand, CDKN2A appears to be significantly downregulated from the stage of oral dysplasia to well- and moderately-differentiated OSCC, with Bcl-2 also declining between the stages of dysplasia and early invasion [33].

It is particularly significant and possibly reflective of why some of those genes play such unique roles in OSCC, the fact that their encoded proteins are involved in co-dependent signaling pathways, and some of them have been found to operate synergistically in OSCC formation. As previously stated, *PTEN*, a tumor suppressor OSCC-related driver gene, negatively regulates the PIK/AKT signaling cascade, which is stimulated by the overexpression of *PIK3CA*, an oncogene also essential for OSCC development [13,52,53]. Furthermore, PTEN loss and Bcl-2 overexpression within the same OSCC tissues have been associated with late-stage tumors, poor differentiation, and lymph node metastasis [76].

Increased EGFR expression, on the other hand, which has been firmly associated with OSCC, activates and drives the overexpression of c-Jun [22], which is also key to OSCC pathogenesis. Finally, the highly implicated tumor suppressor genes *FAT1* and *CASP8* are thought to have synergistic effects on oral oncogenesis, as loss-of-function mutations in both are reported in the majority of OSCC tumors [45].

## 3. Results of Bioinformatic Analysis of miRNA/Target Interactions

The bioinformatic approach used to predict which miRNAs target the two selected custom tumor suppressor gene and oncogene panels yielded 437 and 828 miRNA candidates, respectively, that are predicted to target at least one gene of each respective gene panel. From both developed miRNA/mRNA interaction networks (Figure 1 and Figure 2), we selected the miRNAs that target at least 60% of each gene panel, meaning at least three of the selected tumor suppressor genes, and at least nine of the selected oncogenes, which yielded forty one and four results, respectively. In light of the number targeting at least three out of the tumor suppressor genes and the small size of the panel that explains the large number of targeting miRNA molecules, we decided to only maintain the miRNAs that target more than three tumor suppressor genes (>60%), as a step of additional filtering in the attempt to distinguish the most representative. For the miRNA results that correspond to the larger oncogene cohort of fifteen, we included all four of the miRNAs that are predicted to target and might regulate the expression of at least nine genes (≥60%).

From the final two sets of miRNAs that target the tumor suppressor and oncogene panels, we selected the miRNA molecules that are reported to be significantly upregulated and downregulated, respectively, in the most recent reviews of the available literature (Table 1 and Table 2). Finally, five miRNAs were selected in total (Table 3), with one falling in both categories. More specifically, the miRNAs that are predicted to target more than three (>60%) of the five tumor suppressor genes and are reported to be significantly upregulated in OSCC were miR-155-5p with a target score of 5/5 and miR-34a-5p also with a target score of 5/5. As to the miRNAs that target at least nine (≥60%) of the fifteen selected oncogenes and exhibit significant downregulation in OSCC, our analysis yielded four miRNA molecules: miR-16-5p (Target score: 9/15), miR-1-3p (Target score: 10/15), miR-124-3p (Target score: 12/15) and miR-34a-5p (Target score: 15/15).

### 3.1. MiR-155-5p in OSCC

According to our bioinformatic filtering, miR-155-5p exhibited the highest target score, while it is predicted to target all five of the selected tumor suppressor genes (*TP53*, *CDKN2A*, *FAT1*, *CASP8*, and *PTEN*) that are highly involved in the pathogenesis and clinical features of OSCC, while *TP53*, *FAT1* and *CASP8* belong to the panel of the most frequently mutated genes in OSCC patients (Figure 3). Furthermore, its upregulation in OSCC has been highly reported, indicating a strong oncogenic effect and being associated with numerous aspects of tumor growth and OSCC progression [18].

#### 3.1.1. Expression Patterns in OSCC

MiR-155-5p is significantly overexpressed in OSCC tissues relative to adjacent healthy margins (*p* < 0.0001) and is highly upregulated in HPV-positive compared to HPV-negative tumors, functioning as a marker useful for distinguishing HPV-induced tumors [78]. In OSCC cell lines, there is a significant correlation between heightened levels of miR-155-5p in OSCC cell lines and a notable augmentation in OSCC cell proliferation, colony formation, as well as enhanced invasive and migratory capabilities. Conversely, the suppression of miR-155-5p yields contrasting outcomes [79].

The upregulation of miR-155-5p in OSCC tissues and cell lines has been found to exhibit a significant association with aggressive characteristics, including larger tumor size, advanced stage and grade, lymph node metastasis facilitated by EMT induction, reduced disease-free survival, and unfavorable overall survival, thus highlighting its potential not only as a diagnostic but also as a prognostic tool [24,79,80]. Moreover, there seems to be a positive correlation between elevated levels of miR-155-5p and the emergence of resistance towards the chemotherapeutic agents 5-FU (5-Fluorouracil) and cisplatin, which are commonly employed in the treatment of OSCC [24,72].

#### 3.1.2. Known Target Genes and Affected Pathways

The main target of miR-155 is the FoxO3a member of the FOXO family of transcription factors, which regulates multiple tumor suppressor genes via the FOXO signaling pathway. Consequently, overexpression of miR-155-5p may disrupt the FOXO-induced immunological and cell-cycle regulation. In addition, miR-155-5p harbors targets that are involved in the b-glycal biosynthesis pathway, which is essential for OSCC progression and lymph node metastasis, in which increased glycosylation of molecules such as adhesion-related proteins takes place [30,78]. In addition, by directly targeting the 3′UTR of its mRNA transcript, miR-155-5p inhibits the expression of *ARID2* and is responsible for a substantial decrease in its levels in OSCC. Inhibition of ARID2 expression by miR-155-5p results in a rise in high levels of E-cadherin, vimentin, and snail proteins, indicating that the miR-155-5p/ARID2 axis is essential for promoting tumor growth and mediating OSCC-related EMT and subsequent metastasis [34,79].

Furthermore, the expression of *TP53INP1* and, consequently, its mRNA and protein levels are also downregulated in light of miR-155-5p overexpression in OSCC. TP53INP1 is a well-known tumor suppressor that regulates apoptosis, cell-cycle arrest, and cell migration. The targeting of miR-155-5p has been proposed as a useful treatment strategy aimed at improving the efficacy of OSCC-targeting chemotherapy [81]. Another known target gene of miR-155-5p is *CDKN1B* which regulates 27Kip1 cyclin-dependent kinase inhibitor, which mediates the cell cycle progression at phase G1, and its degradation is required for cell proliferation to occur. While the upregulation of 27Kip1 exerts apoptotic and anticancerous effects, its downregulation has been associated with the progression of numerous neoplasms. In the case of OSCC, the typical upregulation of miR-155-5p reduces p27Kip1 levels, thereby promoting proliferation and oncogenesis, whereas the suppression of miR-155-5p results in the upregulation of p27Kip1 expression, which reduces proliferation and inhibits tumorigenicity [30]. Finally, miR-155-5p has been shown to induce oncogenesis in OSCC by suppressing the expression of CDC73 (parafibromin), a key tumor suppressor, thereby inhibiting apoptosis and promoting growth while simultaneously inducing inflammation by suppressing the expression of SOCS1, a well-known anti-inflammatory factor [24,26].

### 3.2. MiR-16-5p in OSCC

According to our miRNA/target interaction analysis, miR-16-5p yielded to target nine of our custom panel of 15 OSCC-associated key oncogenes (Figure 4). More specifically, the *PIK3CA*, *MYC*, *JUN*, *EGFR*, *FGF2*, *FGFR1*, *FGFR4*, *MKI67,* and *BCL2* genes, which are known to be highly involved in OSCC pathogenesis, are bioinformatically predicted to be targeted by miR-16-5p, which in turn is highly reported to exhibit significant downregulation in OSCC tissues and cell lines [25,30,82,83,84].

#### 3.2.1. Expression Patterns in OSCC

It has been reported that miR-16-5p, also known as miR-16, fulfills a tumor suppressor role in OSCC by inducing apoptosis of malignant cells and inhibiting tumor growth [30,82]. Compared to normal specimens and tumor-free adjacent tissues, OSCC tumors and cell lines exhibit significantly lower expression of miR-16, and it has been shown to be especially downregulated in higher-grade lesions, thus serving as a potential non-invasive tool for OSCC diagnosis and distinguishing advanced tumors [25,30,82,83]. It has been reported that miR-16 is downregulated in approximately 60% of OSCC tumors [84], and it has been strongly correlated with lower disease-free and overall survival rates of patients, highlighting its potential as a prognostic marker [82,85].

The introduction of miR-16-mimicking molecules into OSCC cell lines results in the inhibition of proliferation and robust apoptotic effects, whereas the inhibition and silencing of miR-16 has the exact opposite effect [83]. The quantification of miR-16 in plasma samples from OSCC patients yielded results opposite to those obtained from tissue. In fact, miR-16 levels were higher in plasma samples, which was attributed to the selective release of tumor-suppressing miRNAs by OSCC cells [86].

#### 3.2.2. Known Target Genes and Affected Pathways

MiR-16 acts as a tumor suppressor by decreasing the expression levels of genes that encode factors involved in the PI3K/Akt signaling pathway, such as BCL2, MTOR, CCND1, CCND3, SGK3, and AKT3, which are implicated in cell cycle progression and cell survival, as well as growth and proliferation thus outlining its tumor suppressing, pro-apoptotic role both in vivo and in vitro. Nonetheless, in the typical case of miR-16-5p downregulation, PI3K/Akt oncogenic signaling may be amplified and overactivated, resulting in tumor growth and disease progression [30,86]. AKT3, as well as BCL2-like protein 2 (BCL2L2), are highly expressed in OSCC tissues and cell lines, which exhibit significantly low expression of miR-16, whereas the induced higher expression of miR-16 and the decrease in their protein levels were successful in reducing OSCC proliferation and tumor size [83].

In addition, miR-16 exerts its tumor-suppressing properties by directly targeting the 3′UTR of the mRNA encoding the Tousled-Like Kinase 1 (TLK1), thereby inhibiting its expression. TLK1 interacts strongly with the AKT-interacting protein (AKTIP), and its elevation leads to the overactivation of the PI3K/AKT pathway, which has been identified as a key driver in numerous malignancies, including OSCC [82,87]. In OSCCs with substantially downregulated expression of miR-16-5p, the expression of TLK1 is significantly elevated compared to adjacent normal tissues, indicating its subsequent dysregulation due to miR-16 suppression [82]. Finally, miR-16 inhibits tumor growth in OSCC and induces apoptosis in vivo and in vitro by inhibiting the Wnt/β-catenin signaling pathway, which is highly upregulated in OSCC and accounts for cell fate, proliferation, and migration. Therefore, the induced overexpression of miR-16 has been proposed as a potential alternative treatment for OSCC [88].

### 3.3. MiR-1-3p in OSCC

Based on the developed miRNA/target interaction network analysis, it was observed that miR-1-3p targetted ten of our custom panel of fifteen OSCC-associated key oncogenes (Figure 5). More specifically, the *PIK3CA*, *HRAS*, *MYC*, *JUN*, *EGFR*, *FGF2*, *FGFR2*, *FGFR4*, *MKI67,* and *ETS1* genes, which are known to be highly involved in OSCC pathogenesis, are bioinformatically predicted to be targeted by miR-1-3p, which in turn is highly reported to exhibit significantly low expression in OSCC [17,26,30,89].

#### 3.3.1. Expression Patterns in OSCC

MiR-1-3p, also referred to as miR-1, is recognized as a tumor suppressor miRNA in OSCC. It plays a pivotal role in promoting apoptosis and inhibiting the migration and invasiveness of tumor cells [30,89]. Conversely, the downregulation of miR-1 leads to the activation of these properties, along with enhanced colony formation in OSCC cell lines [26,30,89]. In OSCC tumors, it is commonly observed that the expression levels of miR-1-3p are markedly diminished. This reduction in miR-1-3p levels has been found to facilitate cancer cell migration and stimulate invasion by activating the process of EMT. This finding elucidates a significant correlation between the notably diminished expression of miR-1 and the occurrence of lymph node metastasis, advanced tumor stages (III and IV), as well as a generally inferior prognosis [17,25,89,90]. In contrast, the upregulation of miR-1 through induction has been shown to possess therapeutic promise in the context of OSCC by effectively suppressing the proliferation and migration of malignant cells [89,90].

#### 3.3.2. Known Target Genes and Affected Pathways

MiR-1 normally suppresses migration and invasion by negatively regulating the expression of Slug or SNAI2 (snail family transcriptional repressor 2), a crucial EMT regulator, by targeting the 3′UTR of its mRNA. In the typical case of miR-1-3p downregulation in OSCC, Slug is overexpressed, resulting in the diminishment of E-cadherin expression, thus promoting EMT and bestowing invasion dynamics to OSCC cells [89].

The *EGFR* gene, which exhibits notably elevated expression in OSCC cells, is identified as an additional direct target of miR-1, further supporting its role in tumor suppression. The downregulation of *EGFR* expression and signaling has been observed in cases where miR-1 overexpression is induced. This downregulation leads to a decrease in the aggressiveness of OSCC and suggests that miR-1 might serve as a potential therapeutic agent [89,91]. The *c-MET* gene, which encodes a tyrosine kinase that plays a role in cellular proliferation, migration, and invasion, has been identified as a direct target of the tumor suppressor miR-1-3p [91,92]. The expression of *c-MET* is observed to be markedly elevated in head and neck SCC, which may be attributed to the substantial decrease in the expression levels of its regulatory molecule miR-1-3p, thus contributing to the activation of oncogenic signaling pathways [91].

Alongside normally regulating migration and invasion, miR-1-3p also inhibits OSCC proliferation by suppressing the expression of the *DKK1* (dickkopf WNT signaling pathway inhibitor 1) gene of the WNT signaling pathway, that is typically overexpressed in various cancers, including OSCC tissues and cell lines, and stimulates proliferation, migration, and invasion. Introducing a miR-1-3p mimic to OSCC cell lines leads to the suppression of migratory and invasive dynamics and to declined proliferation by inducing the suppression of *DKK1* expression levels [90].

### 3.4. MiR-124-3p in OSCC

According to the miRNA/target interaction analysis in silico, miR-124-3p interacts with 12 of our custom panel of 15 OSCC-associated oncogenes (Figure 6). More specifically, the mRNAs of the *NOTCH1*, *HRAS*, *PIK3CA*, *EGFR*, *ERBB2*, *FGFR1*, *FGFR3*, *FGFR4*, *ETS1*, *JUN*, *MKI67*, and *MYC* genes, which play a key role in the development and clinicopathological features of OSCC, are predicted to be targeted by miR-124-3p, which in turn is notably decreased in OSCC tissues and cell lines in terms of expression [25,31].

#### 3.4.1. Expression Patterns in OSCC

MiR-124-3p, also referred to as miR-24, is recognized as a tumor suppressor miRNA in a number of cancers, including OSCC. It possesses the capacity to impede the adhesion and movement of malignant cells, induce programmed cell death, and hinder tumor growth [25,31,93]. The expression levels of miR-124-3p have been observed to exhibit a notable downregulation in both tissue and saliva samples obtained from patients diagnosed with OSCC [25,31,94]. Additionally, animal models of OSCC have demonstrated a similar decrease in miR-124-3p levels within tumor cells [95]. Upregulation of miR-124-3p in OSCC cell lines has been shown to be a robust indication of its ability to suppress tumor growth through the inhibition of cancer cell migration and invasion [93].

The downregulation of miR-124-3p is a distinguishing feature observed in SCCs affecting the oropharynx and oral cavity. Conversely, it has been documented to be upregulated in SCCs affecting the larynx and pharynx. In addition, the expression patterns of miR-124-3p have the potential to be utilized in distinguishing between HPV-positive and HPV-negative OSCCs. This is due to its notably reduced expression in the presence of HPV, while it is upregulated in HPV-free tumors. In addition, it is worth noting that miR-124-3p may potentially function as a molecular marker for staging OSCC. This is supported by the significant downregulation of miR-124 in stage IV OSCCs, as opposed to the upregulation observed in stages II and III [94]. In addition, the large downregulation of miR-124-3p has been observed to confer resistance to cisplatin, thereby posing a significant challenge to the efficacy of therapeutic interventions against OSCC [96].

#### 3.4.2. Known Target Genes and Affected Pathways

MiR-124 has been documented to selectively target multiple genes that have been implicated in the development and progression of malignancies, thereby exerting its suppressive effects on tumor growth. However, these tumor suppressive effects are compromised when miR-124 is downregulated in OSCC [93,95,96,97]. MiR-124 is reported to exert a suppressive effect on OSCC motility. This is achieved through the targeting of the mRNA of the *ITGB1* gene, which encodes the integrin subunit β 1 (ITGB1). The ITGB1 protein is known to play a crucial role in the oncogenic PI3K/AKT cascade. MiR-124 typically decreases the levels of ITGB1 protein and mRNA expression in OSCC cells through its interaction with two conserved binding sites located in the 3′UTR of its mRNA. The results of miR-124 in OSCC cells support its tumor-suppressive role. This is evidenced by the significant decrease in ITGB1 expression within OSCC cells following miR-124-induced overexpression. Additionally, miR-124 intervention leads to the inhibition of OSCC cell adherence and motility. These findings suggest that dysregulation of miR-124 may play a critical role in promoting the progression of OSCC [93].

The gene encoding CCL2, also known as monocyte chemoattractant protein-1 (MCP-1), is another target of miR-124-3p that is highly correlated with cancer cell migration and overall malignant progression in general. CCL2 is frequently upregulated in OSCC tissues and is also observed to be overexpressed in plasma samples of OSCC patients. The overexpression of CCL2 has been found to facilitate tumor progression by attracting immune cells to the tumor microenvironment, with the ability to secrete a diverse array of growth factors and cytokines, thereby augmenting both the growth and invasion capabilities of the malignancy. The observed decrease in expression of miR-124-3p, leading to its diminished tumor-suppressive function, may provide a potential explanation for the observed increase in CCL2 levels [95,97]. Finally, it is indicated that miR-124 suppresses the expression of TRIM14 (Tripartite Motif Containing 14) by interacting with the 3′UTR of its mRNA. The dysregulation of TRIM14 has been observed in various malignancies and has been shown to facilitate cell proliferation and inhibit apoptosis in colorectal cancer by suppressing the PTEN tumor suppressor [96,98]. The upregulation of TRIM14 expression, potentially caused by the downregulation of miR-124, has been documented as a contributing factor in the development of tongue OSCC. Additionally, this upregulation has been found to confer chemoresistance to cisplatin, a commonly used treatment for OSCC. In accordance with the above, the overexpression of miR-124 is observed to significantly downregulate TRIM14 levels while also reducing cisplatin resistance in tongue OSCC [96].

### 3.5. MiR-34a-5p in OSCC

Based on our computational analysis of miRNA/target interactions, miR-34a-5p demonstrated the highest target score among all miRNAs in both custom gene panels, achieving a perfect score of 20 out of 20. MiR-34a-5p has been identified as a potential regulator of all fifteen oncogenes (*NOTCH1*, *HRAS*, *PIK3CA*, *EGFR*, *ERBB2*, *FGFR1*, *FGFR2*, *FGFR3*, *FGFR4*, *FGF2*, *ETS1*, *JUN*, *MKI67*, *MYC*, *and BCL2*) and all five tumor suppressor genes (*TP53*, *CDKN2A*, *FAT1*, *CASP8*, *and PTEN*) that are known to play crucial roles in the development of OSCC (Figure 7 and Figure 8). This miRNA, which is known for its role in cancer development, stands out as the most OSCC-specific molecule among all the candidates that have successfully passed the target-score filtering in our analysis. MiR-34a-5p appears to exhibit a multifaceted role in OSCC, as it not only encompasses two corresponding gene categories but has also been documented to exhibit both downregulation and upregulation in OSCC biological specimens, as reported in the current literature [27,32,99,100].

#### 3.5.1. Expression Patterns in OSCC

MiR-34a-5p, alternatively referred to as miR-34a, has been identified as a well-established tumor suppressor in various types of cancer [99]. In regards to OSCC, research has predominately revealed a major decline in the levels of miR-34a-5p in OSCC tumor specimens and OSCC cells when compared to normal tissues and cell lines. This reduction has exhibited a strong association with more aggressive phenotypes, as well as with lymph node metastasis and unfavorable overall prognosis among OSCC patients [27,32,100]. In contrast, the exogenous stimulation of miR-34a in OSCC cell lines leads to the inhibition of EMT and a notable reduction in the cells’ ability to invade and migrate. In addition, the overexpression of miR-34a causes the arrest of cells in the G1 phase of the cell cycle, thereby inhibiting the growth and proliferation of OSCC. This highlights the tumor-suppressive function of miR-34a, as it not only hinders tumor growth but also provides protection against cell migration and metastasis [100].

The downregulation of miR-34a is observed in precancerous oral conditions as well. This is evidenced by the significantly reduced levels of miR-34a in saliva samples obtained from patients with leukoplakia compared to samples from corresponding healthy controls. These findings highlight the potential implications of miR-34a in the early stages of oral oncogenesis, as well as into malignant transformation [27]. In addition, it has been observed that the expression levels of miR-34a-5p are markedly reduced in exosomes originating from cancer-associated fibroblasts (CAFs) in OSCC. CAFs play a critical role in promoting malignant progression by releasing exosomes that contain various epigenetic factors, particularly miRNA molecules, which are acquired from neighboring cells. The observed decrease in the expression of miR-34a in OSCC CAFs has been found to be strongly associated with increased cell proliferation and a notable rise in the metastatic capacity of the malignant tumor. In contrast, the upregulation of miR-34a-5p in OSCC CAFs exhibits contrasting outcomes, as it inhibits the progression of OSCC and modulates the tumor’s malignant behavior [101].

Contrary to prevailing consensus, a subset of studies has characterized miR-34a as an oncomiR in OSCC, positing its involvement in the pathogenesis and progression of the neoplasm through the facilitation of malignant proliferation [27]. However, these findings have stimulated ongoing debate and controversy. Indeed, a number of studies have demonstrated that miR-34a exhibits increased expression in OSCC tissues in comparison to normal oral specimens. Additionally, elevated levels of the miRNA have been detected in saliva samples from individuals diagnosed with OSCC, leading to its potential application as a non-invasive method for detecting OSCC and aiding in the early diagnosis of cancerous lesions [25,27]. Pertaining to miRNA expression patterns in head and neck SCCs, overexpression of miR-34a has been primarily reported in tissue specimens of laryngeal cancer, as opposed to oral cancerous tissues. However, miR-34a-5p upregulation in both OSCC and laryngeal cancer specimens has been correlated with a better prognosis and lower mortality rates. This further emphasizes the protective effects of miR-34a-5p rather than its oncogenic potential [102].

#### 3.5.2. Known Target Genes and Affected Pathways

Interleukin 6 receptor (IL6R), which activates the oncogenic STAT3 transcription factor, is a verified direct target of miR-34a that interacts with the 3′UTR of its mRNA [100,103]. It is demonstrated that miR-34a is capable of modulating the intrinsic expression of IL6R in OSCC cells. This is supported by the observation that the mRNA levels of IL6R are significantly decreased in cell lines where miR-34a overexpression has been exogenously induced. In contrast, it has been reported that OSCC cells with diminished expression of miR-34a demonstrate significant upregulation of the receptor. The upregulation of miR-34a leads to the downregulation of the IL6/STAT3 signaling pathway, primarily mediated by IL6R. Both IL6R and STAT3 levels experience a substantial reduction when miR-34a is upregulated. It is hypothesized that this phenomenon is one of the underlying mechanisms through which miR-34a suppresses proliferation and metastasis. Consequently, the overexpression of miR-34a has been suggested as a potentially effective therapeutic strategy for combating OSCC. The downregulation of miR-34a in OSCC has been observed to have contrasting effects, potentially leading to an upregulation of IL6/STAT3 oncogenic signaling. This, in turn, may contribute to the exacerbation of malignant progression and metastasis in OSCC [100].

Matrix Metalloproteases 9 and 14 (MMP9, MMP14), the expression of which is known to be significantly upregulated in all cancer types, are two additional validated target genes of miR-34a, with binding sites lying on the 3′UTRs of their mRNAs [99,104]. The levels of endogenous proteins MMP9 and MMP14 were found to be significantly decreased due to the overexpression of miR-34a in tongue OSCC cell lines. This reduction in protein levels was also observed to be associated with the inhibition of invasion and migration. In contrast, when miR-34a was suppressed, the expression levels of both metalloproteases exhibited significant increases [32,99]. Furthermore, there are reports indicating that miR-34a-5p plays a role in regulating the expression levels of its direct target gene *AXL*, with the aim of inhibiting the proliferation and metastasis of OSCC. The activation of *AXL* signaling has been linked to increased survival, proliferation, migration, and invasion of cancer cells [101,105]. In the context of OSCC, the overexpression of AXL has been associated with elevated rates of proliferation and migration. This may be attributed to the downregulation of miR-34a, a frequent finding in OSCC, which promotes tumor progression, EMT, and metastasis. These effects are mediated through the overactivation of the AKT/GSK-3β/β-catenin/Snail signaling pathway. Consequently, this pathway upregulates the expression of MMP2 and MMP9, leading to increased aggressiveness and disease progression in OSCC [101].

Finally, the *SATB2* (special AT-rich binding protein-2) oncogene, known for its significant involvement in cancer development, is directly targeted by miR-34a. The downregulation of the miR-34a/SATB2 axis has a significant functional impact on the growth, invasion, and migration of OSCC cells. In particular, it was observed that OSCC tissues with reduced levels of miR-34a demonstrate a significant increase in the expression of SATB2. On the contrary, the overexpression of miR-34a had a tumor-suppressive effect by causing a notable decrease in SATB2 expression, thereby inhibiting the proliferation, invasion, and migration associated with OSCC [106,107].

In summary, while a handful of studies have identified miR-34a as an oncogenic molecule, the majority of relevant research consistently emphasizes its significant tumor suppressive function in OSCC. Nevertheless, it is imperative to acknowledge the contentiousness of the expressional findings, as they may indicate a potential duality in its role in oral oncogenesis.

## 4. Discussion

Oral cancer, a significant public health concern, encompasses a diverse group of malignancies affecting the oral cavity and oropharynx, represented in most cases by OSCC. Despite advancements in treatment modalities, the prognosis for oral cancer remains suboptimal, emphasizing the need for a better understanding of its molecular mechanisms [8,11]. The genetic basis of OSCC is widely acknowledged and extensively recognized, with key genes playing crucial roles in its development and progression. Tumor suppressor genes, including *TP53*, *CDKN2A*, *FAT1*, and *CASP8*, are frequently altered or inactivated in OSCC, resulting in disrupted cell cycle regulation, enhanced cell survival, and impaired apoptosis. Conversely, oncogenes such *as NOTCH1*, *HRAS*, *PIK3CA*, *EGFR*, *ERBB2*, *FGFR1-4*, *FGF2*, *ETS1*, *JUN*, *MKI67*, *MYC*, and *BCL2* are often overexpressed or harbor activating mutations, driving increased cell proliferation, invasion, and resistance to cell death [14,19,20,21,22,75,77].

While the genetic landscape of OSCC is well-established, in recent years, the focus has turned to epigenetic elements that may be implicated in its pathogenesis and progression, such as miRNAs, which are crucial regulators of gene expression and have the potential to be used as non-invasive diagnostic and prognostic biomarkers, since they can be readily detected in body fluids, paving the way for the development of sensitive and specific tests for OSCC [12,23,102]. Increased research concerning the expression patterns of miRNAs in OSCC during the past few years has generated a vast number of observations, posing challenges in terms of their interpretation. In a general context, the mechanisms underlying cancer development demonstrate significant overlap, leading to the emergence of common patterns in miRNA expression across different types of malignancies being studied. As a result, we are faced with a multitude of findings that hold statistical significance. Nevertheless, it is extremely challenging to definitively assert that these findings truly reflect OSCC to such a significant degree that they possess the potential to unveil distinctive underlying mechanisms or serve as dependable molecular tests for this specific neoplasm.

Hence, in order to elucidate the most important among this extensive array of observed molecules implicated in the pathogenesis of OSCC, we opted to strategically discern the most crucial and illustrative miRNA molecules associated with this particular malignancy by leveraging the inherent relationship between genetics and epigenetics, focusing on the key genes implicated in every phase of oral oncogenesis. After conducting an in-depth examination of the existing knowledge on miRNA expression in OSCC, we have carefully selected a set of 20 key cell-cycle regulatory genes that have been widely recognized for their significant involvement in OSCC pathogenesis. These genes have been divided into two distinct custom panels, the first comprising fifteen oncogenes and the second five tumor suppressor genes (Figure 9).

The two custom panels were utilized for the identification of all miRNA molecules that are predicted by in silico analysis to target the selected genes. We constructed miRNA/target interaction networks for both oncogene and tumor suppressor panels, and from a vast number of miRNA molecules, we have identified a subset of miRNAs that are predicted to target a minimum of 60% of the fifteen oncogenes as well as at least 60% of the panel of five tumor suppressor genes. The miRNAs that exhibited a significant upregulation in OSCC based on research findings and were predicted to target over 60% of the tumor suppressor gene panel include miR-155-5p and miR-34a-5p with major target scores of 5/5. In turn, the miRNAs that target at least 60% of the oncogene panel and demonstrate significant downregulation in OSCC were miR-16-5p, miR-1-3p, miR-124-3p, and miR-34a-5p with a target score of 9/15, 10/15, 12/15 and 15/15, respectively (Figure 9).

All the miRNAs we portrayed as important and OSCC-specific have been studied in the past, and some of the signaling pathways which are affected by their dysregulation in OSCC have been elucidated. Nevertheless, their regulatory effects on the mRNA and protein expression levels of the factors we included in our analysis remain unexplored. In contrast, the existing research literature has primarily examined alternative gene targets that are not encompassed within our customized gene panels specific to OSCC but might affect some of the signaling pathways they are involved in, such as the oncogenic PI3K/AKT cascade. In this review, the current understanding of the expression patterns of important OSCC-implicated miRNAs has been thoroughly discussed.

MiR-155-5p, which targets our entire custom OSCC-related tumor suppressor panel, is overexpressed in OSCC tissues and cell lines and has been associated with tumor growth, aggressive OSCC phenotypes, EMT, as well as with lymph node metastasis. The overexpression of miR-155 appears to significantly activate oncogenic signaling pathways by downregulating critical tumor suppressor genes such as *FoxO3a*, *ARID2*, *TP53INP1*, *CDKN1B*, and *CDC73*, causing cell cycle dysregulation and apoptosis inhibition [25,30,43,83,84,85].

MiR-16-5p, which is anticipated to target nine genes from our distinct panel of fifteen OSCC-associated vital oncogenes, is downregulated in the majority of OSCCs and is considered to be a tumor suppressor. Reduced miR-16-5p expression suppresses apoptosis and promotes tumor growth, thus rendering it a reliable prognostic marker for OSCC [25,30,43,83,84,85]. MiR-16 targets multiple genes involved in cell cycle regulation, apoptosis, and metastasis, including *BCL2*, *BCL2L2*, *MTOR*, *CCND1*, *CCND3*, *SGK3*, *AKT3*, and *TLK1*. Its downregulation amplifies the oncogenic PI3K/AKT and Wnt/β-catenin signaling pathways, which have been implicated in many malignancies, including OSCC [25,30,43,83,84,85,87,101].

MiR-1-3p targets 12 of the 15 oncogene genes and is typically downregulated in OSCC, thus promoting cell growth and leading to the inhibition of apoptotic processes. It is known that miR-1-3p targets *EGFR*, *c-MET*, and *DKK1* gene transcripts, which are overexpressed and accelerate OSCC progression in cases of miR-1 downregulation [89,90,91,92]. Overexpression of Slug, another miR-1-3p target, promotes OSCC EMT and invasion following miR-1 downregulation [89].

MiR-124-3p has been identified as a tumor suppressor molecule in numerous types of cancer, including OSCC, and is predicted to target 10 of our 15 oncogene panel genes according to our computational analysis. Tissue and saliva samples from patients reveal significant downregulation of this miRNA in OSCC [25,31,94]. In OSCC animal models, miR-124-3p levels exhibit a significant decrease within tumor cells [95], while their restoration declines cell proliferation, migration, and invasion [93]. MiR-124-3p exerts tumor-suppressive effects by targeting several oncogenes, such as *ITGB1*, *TRIM14*, and *CCL2*, which are overexpressed under miR-124 downregulation, subsequently stimulating oncogenic pathways that include the PI3K/AKT cascade, intense growth factor signaling, or the downregulation of PTEN tumor suppressor in the case of TRIM14 upregulation. These diminish miR-124-induced tumor suppression and promote tumor growth and OSCC progression in turn [93,96,97,101].

Finally, miR-34a-5p scored a perfect 20 in both gene panels. It is predicted to target and may regulate all fifteen oncogenes (*NOTCH1*, *HRAS*, *PIK3CA*, *EGFR*, *ERBB2*, *FGFR1*, *FGFR2*, *FGFR3*, and *FGFR4*) and all five tumor suppressor genes (*TP53*, *CDKN2A*, *FAT1*, *CASP8*, and *PTEN*), which are signature genes in oral oncogenesis. In multiple types of cancer, miR-34a-5p regulates apoptosis, cell cycle, and cellular senescence. OSCC tumor specimens, cells, and CAF exosomes exhibit substantially lower miR-34a-5p levels than normal tissues and cell lines. Several studies have shown that this decline is strongly associated with aggressive OSCC characteristics, lymph node metastases, and poor patient prognosis [27,32,100,101]. MiR-34a downregulation in saliva samples from leukoplakia patients also suggests its role in early oral carcinogenesis [27].

It is worth mentioning that, contrary to consensus, a subset of studies has referred to miR-34a as an oncogenic factor in OSCC, positing that its upregulation in OSCC tissues facilitates malignant proliferation and contributes to the pathogenesis and progression of the neoplasm [27]. However, miR-34a-5p upregulation in OSCC cases has been correlated with a better prognosis and lower mortality rates, supporting its protective rather than oncogenic involvement in this particular malignancy [102]. According to the available research findings, miR-34a-5p suppresses tumor growth by targeting *IL6R*, *MMP9*, *MMP14*, *AXL*, and *SATB2* genes, which are upregulated in the typical case of miR-34a suppression in OSCC, therefore overactivating several oncogenic signaling pathways including the IL6/STAT3 and AKT/GSK-3β/β-catenin/Snail cascades [99,100,101].

In summary, our findings suggest that miR-34a-5p, miR-155-5p, miR-124-3p, and miR-16-5p are the most representative of OSCC from a large pool of over 1000 molecules appearing to be associated with OSCC pathogenesis and characteristics. The OSCC-associated predicted targets of these molecules, which have not been explored yet in terms of expression assessment alongside those miRNAs, might hold the explanation for why they experimentally exhibit such typical and consistent expressional patterns in OSCC. However, that notion remains to be investigated. Only in the case of miR-16-5p one of our selected oncogenes, *BCL2,* has been experimentally verified as a target, providing a partial explanation of the tumor suppressive potential of this miRNA [84]. The experimental validation of this specific target aligns with our in silico analysis results and is indicative that the rest of our strategical computational predictions might potentially be experimentally verified as well, providing evidence that these kinds of sequence-based predictions are worthy of further investigation.

The results of this in silico analysis revealed the roles of miR-34a-5p, miR-155-5p, miR-124-3p, miR-1-3p, and miR-16-5p in oral cancer and may provide the basis for additional research to take place in a yet unexplored territory. Studying the expression levels of the five miRNAs in relation to the expression levels of major oncogenes and tumor suppressor genes in OSCC specimens compared to normal tissues holds tremendous potential for further research. The comprehensive understanding acquired through investigating the collective expression patterns of these miRNAs and their target genes, which have the highest association with this particular disease, may pave the way for advancements in the diagnosis, prognosis, and personalized future treatment approaches for OSCC.

The benefits of this strategy extend beyond OSCC and have the potential to contribute to cancer research as a whole. By elucidating the complex interplay between miRNAs and their target genes that are characteristically involved in a specific malignancy or pathology in general, it is plausible to unveil disease-specific regulatory mechanisms, as well as to critically assess the extensive corpus of available relevant miRNA expression data, leading hopefully to the identification of the characteristic epigenetic signatures of each disease in the future. Hence, this particular strategy has the potential to provide valuable guidance in the design and advancement of novel therapeutic methodologies, such as miRNA-based therapeutics, which exhibit significant promise in the field of precision medicine.

## Figures and Tables

**Figure 1 genes-14-01578-f001:**
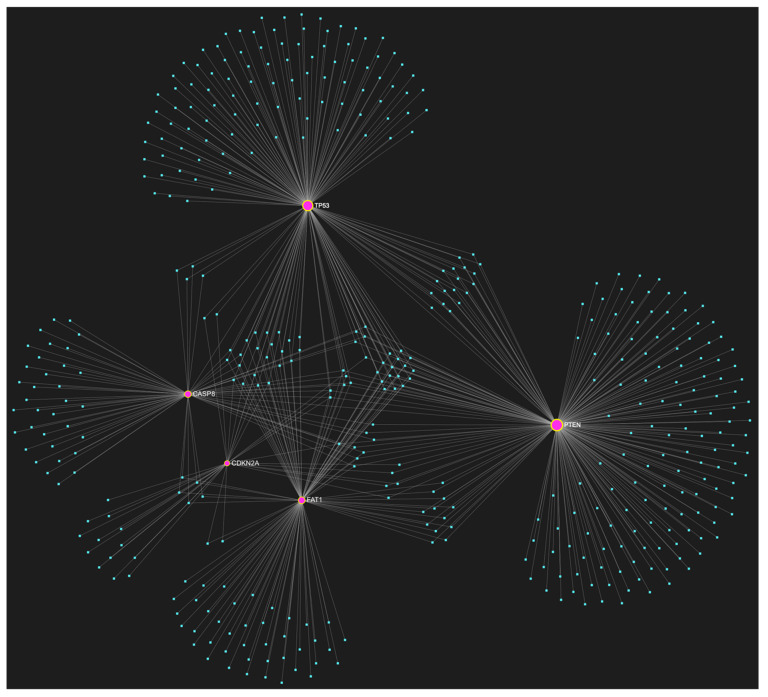
The miRNA/target interaction network illustrates the 437 miRNA molecules that are predicted to target one or more genes within our custom panel of tumor suppressor genes specific to OSCC, namely *TP53*, *CDKN2A*, *FAT1*, *CASP8*, and *PTEN*. The genes encompassing the panel are visually represented by the color pink, whereas the miRNA molecules that are anticipated to target at least one of these genes are visualized by the color blue.

**Figure 2 genes-14-01578-f002:**
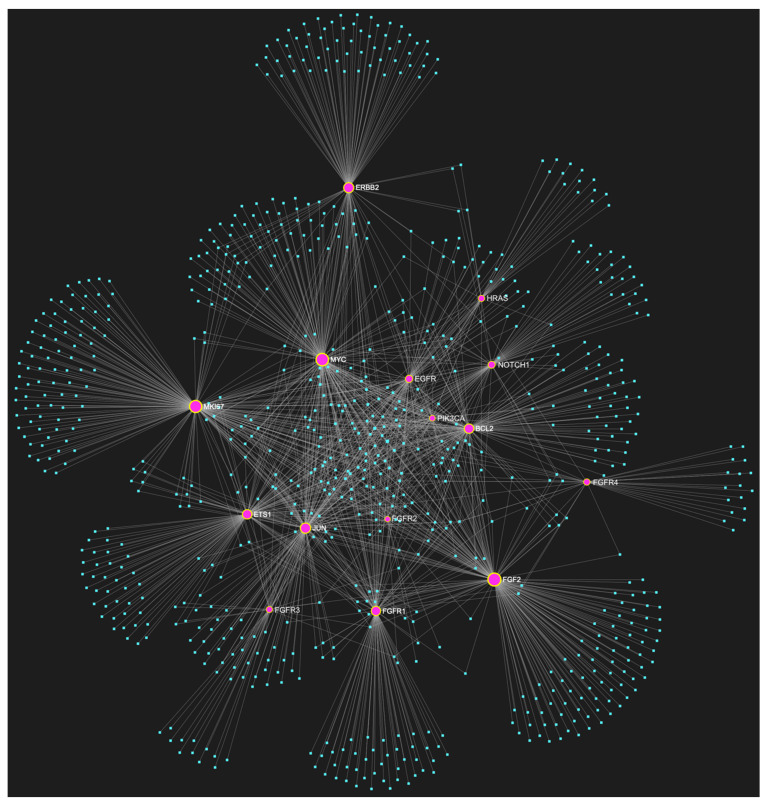
The miRNA/target interaction network illustrates the 828 miRNA molecules that are predicted to target at least one gene within our custom panel of oncogenes specific to OSCC. The oncogenes included in the panel are *NOTCH1*, *HRAS*, *PIK3CA*, *EGFR*, *ERBB2*, *FGFR1*, *FGFR2*, *FGFR3*, *FGFR4*, *FGF2*, *ETS1*, *JUN*, *MKI67*, *MYC*, and *BCL2*. The genes encompassing the panel are visually represented by the color pink, whereas the miRNA molecules that are anticipated to target at least one of these genes are visualized by the color blue.

**Figure 3 genes-14-01578-f003:**
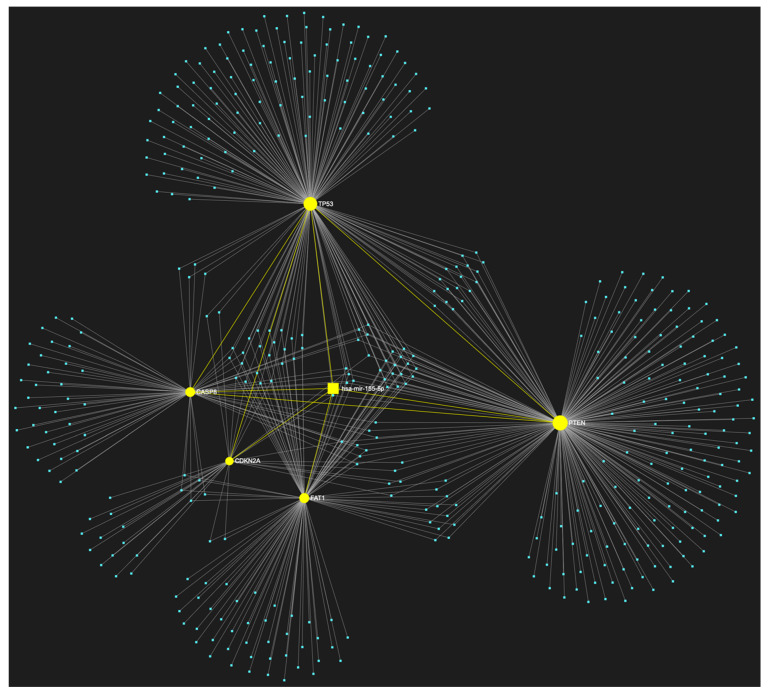
MiR-155-5p, a well-documented upregulated microRNA in OSCC, is predicted to target all five tumor suppressor genes included in our custom OSCC-specific tumor suppressor gene panel (*TP53*, *CDKN2A*, *FAT1*, *CASP8*, *PTEN*). The depicted elements in yellow represent the genes that are specifically targeted by miR-155-5p, along with their corresponding connecting nodes. The blue dots in the illustration correspond to the remainder of miRNA molecules that are anticipated to target at least one gene within this network.

**Figure 4 genes-14-01578-f004:**
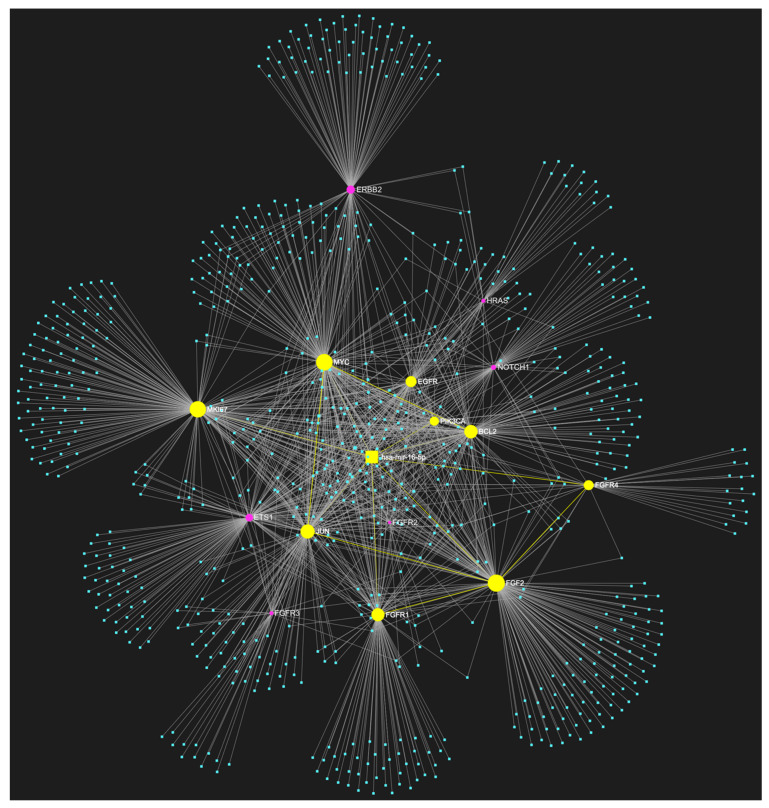
The miRNA/target interaction network illustrates the anticipated targets of miR-16-5p within our customized panel of oncogenes which are highly associated with OSCC. MiR-16-5p has been predicted to target a total of nine oncogenes out of an ensemble of fifteen. These oncogenes include *PIK3CA*, *MYC*, *JUN*, *EGFR*, *FGF2*, *FGFR1*, *FGFR4*, *MKI67*, and *BCL2*. It is worth noting that miR-16-5p has consistently been reported to be downregulated in oral OSCC. The genes targeted by miR-16-5p and their corresponding connecting nodes are depicted in yellow, while genes that are not targeted by this miRNA are visualized in pink. The blue dots in the illustration correspond to the remainder of miRNA molecules that are anticipated to target at least one gene within this network.

**Figure 5 genes-14-01578-f005:**
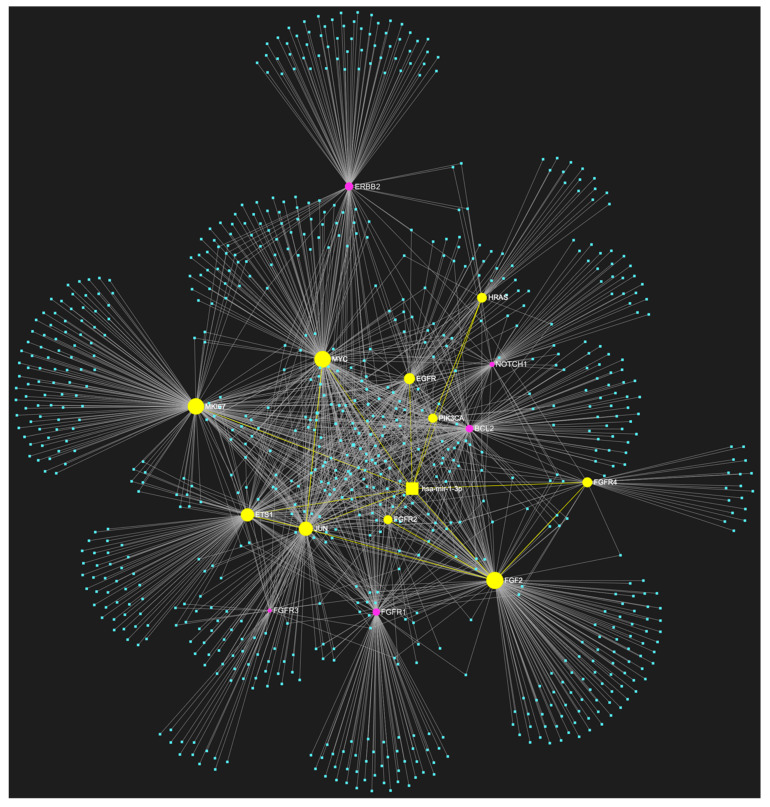
The miRNA/target interaction network illustrates the ten predicted targets of miR-1-3p derived from the focused panel consisting of fifteen key oncogenes associated with OSCC. In particular, it is predicted that miR-1-3p, which is significantly downregulated in OSCC, may target and potentially regulate the expression of *PIK3CA*, *HRAS*, *MYC*, *JUN*, *EGFR*, *FGF2*, *FGFR2*, *FGFR4*, *MKI67* and *ETS1*. These genes are well known for their significant involvement in the pathogenesis of OSCC and are visually represented in yellow, alongside their corresponding connecting nodes, while genes that are not targeted by this miRNA are visualized in pink. The blue dots in the illustration correspond to the remainder of miRNA molecules that are anticipated to target at least one gene within this network.

**Figure 6 genes-14-01578-f006:**
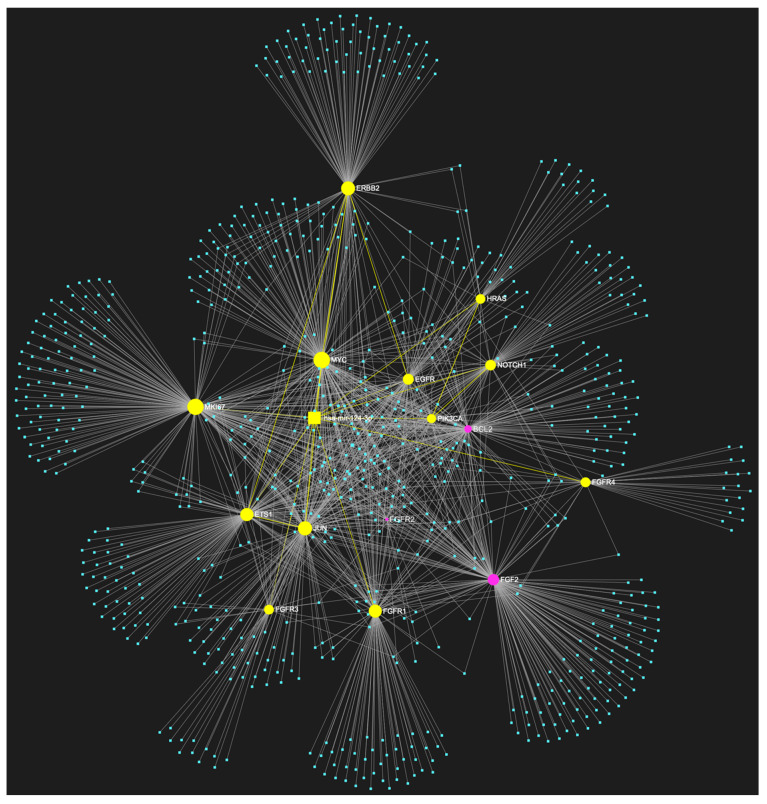
The presented miRNA/target interaction network depicts the expected targets of miR-124-3p within the customized selected panel of oncogenes that exhibit a strong correlation with OSCC. miR-124-3p has been computationally predicted to exhibit targeting potential towards a total of 12 oncogenes from the set of 15. The oncogenes encompassed in this list comprise *PIK3CA*, *NOTCH1*, *HRAS*, *MYC*, *JUN*, *EGFR*, *FGFR1*, *FGFR3*, *FGFR4*, *MKI67*, and *ETS1*. It is noteworthy to mention that miR-124-5p has consistently exhibited downregulation in OSCC. The genes that are subject to targeting a possible regulation by miR-124-3p and their associated nodes are represented in yellow, whereas genes that are not targeted by this specific miRNA are visually represented in pink. The blue dots in the illustration correspond to the remainder of miRNA molecules that are anticipated to target at least one gene within this network.

**Figure 7 genes-14-01578-f007:**
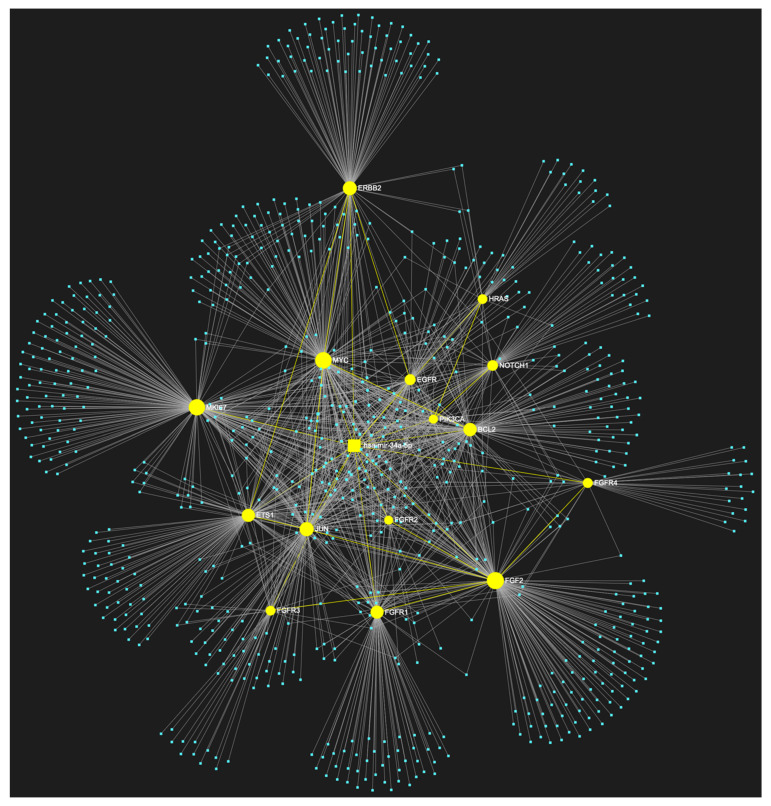
MiR-34a-5p, which has primarily been reported as downregulated but has also been reported by some studies to be upregulated in OSCC, is predicted to target all 15 oncogenes (*NOTCH1*, *HRAS*, *PIK3CA*, *EGFR*, *ERBB2*, *FGFR1*, *FGFR2*, *FGFR3*, *FGFR4*, *FGF2*, *ETS1*, *JUN*, *MKI67*, *MYC*, and *BCL2*) comprising our custom OSCC-specific oncogene panel. The visual elements portrayed in yellow represent the genes that are specifically targeted by miR-34a-5p alongside their corresponding connecting nodes. The blue dots in the illustration correspond to the remainder of miRNA molecules that are anticipated to target at least one gene within this network.

**Figure 8 genes-14-01578-f008:**
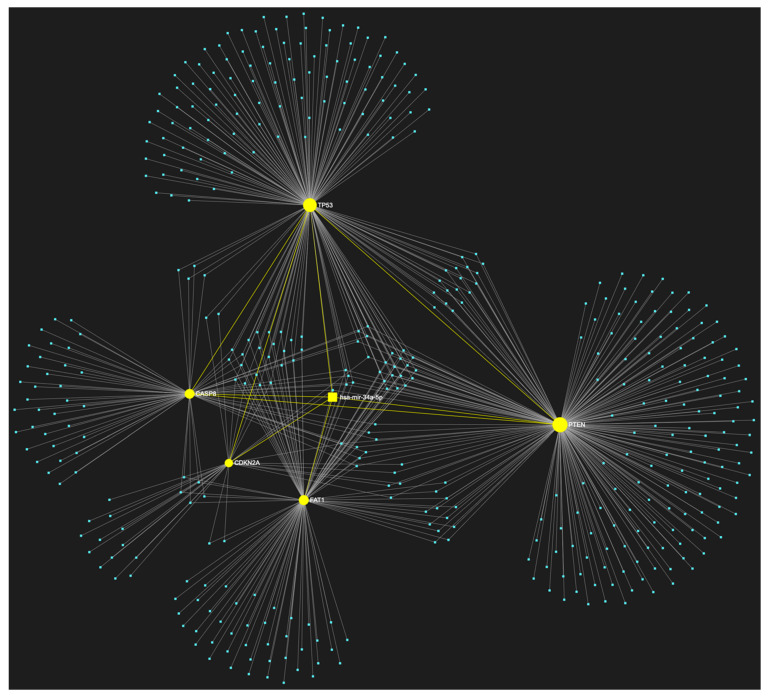
MiR-34a-5p, which has primarily been reported as downregulated but has also been reported by some studies to be upregulated in oral squamous cell carcinoma (OSCC), is predicted to target all five tumor suppressor genes (*TP53*, *CDKN2A*, *FAT1*, *CASP8*, *PTEN*) included in our custom OSCC-specific tumor suppressor gene panel. The visual elements portrayed in yellow represent the genes that are specifically targeted by miR-34a-5p alongside their corresponding connecting nodes. The blue dots in the illustration correspond to the remainder of miRNA molecules that are anticipated to target at least one gene within this network.

**Figure 9 genes-14-01578-f009:**
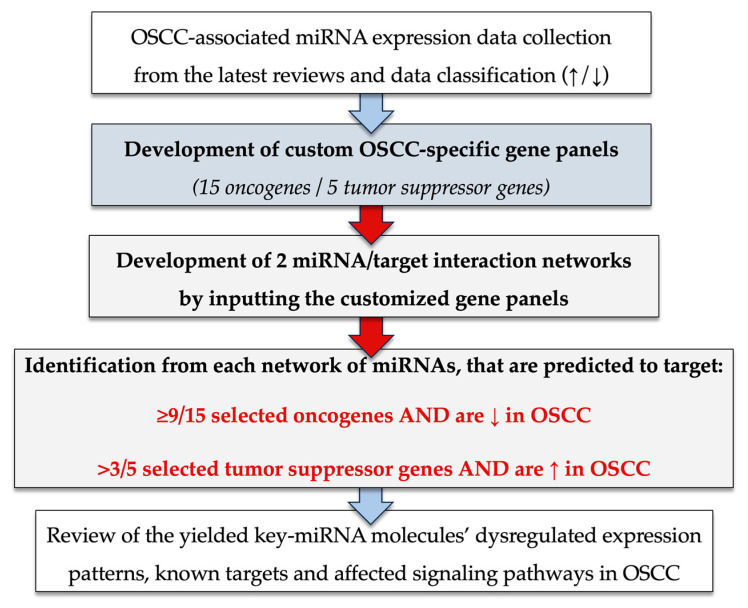
Concise overview flowchart of the methodology employed in this review. The flowchart outlines the various steps involved in our in silico analysis and the criteria used for the bioinformatic filtering of the miRNAs that were revealed from the original miRNA/target gene interaction networks. We propose this bioinformatic strategy as a novel approach for selecting the most crucial miRNAs from a wide range of molecules that have been associated with a certain condition. This approach might hold potential applicability to various pathologies with established genetic backgrounds. The red arrows depicted in the flowchart symbolize the sequential order of the steps involved in the conducted bioinformatic analysis, while the blue arrows indicate the steps that incorporated literature research or review. The red text presents the criteria that were applied to filter the results and ultimately identify the miRNAs that are the most specific to OSCC.

**Table 1 genes-14-01578-t001:** MicroRNA (miRNA) molecules that have been documented to exhibit increased expression levels in biological samples associated with oral squamous cell carcinoma (OSCC), such as tumor tissue, OSCC cell lines, saliva, whole blood, plasma, and serum, compared to non-OSCC biological materials.

↑ miRNA	Sample Source	↑ miRNA	Sample Source
let-7a-3p	Tissue [17]	miR-222	Tissue, Cell lines [17,19,24]
let-7i	Tissue [17]	miR-223	Tissue, Plasma, Serum [17,18,25]
miR-106b	Cell lines [17]	miR-24-3p	Tissue, Saliva, Plasma, Serum [14,17,18,26,27,28]
miR-10a	Tissue, Cell lines [25,29]	miR-25	Serum [17]
miR-10b	Tissue, Cell lines, Plasma [14,25]	miR-26a	Tissue, Cell lines [19,25]
miR-117	Tissue, Cell lines [19]		
miR-118	Tissue, Cell lines [19]		
miR-1246	Tissue, Cell lines, Salivary exosomes [17,25,28]	miR-27a-3p	Tissue, Cell lines [18,24,25,30]
miR-1250	Saliva [18]	miR-27b	Tisuue, Cell lines, Saliva [28]
miR-1269a	Tissue, Cell lines [19]	miR-29b	Tissue, Cell lines [17]
miR-127	Tissue [17]	miR-31-5p	Tissue, Cell lines, Saliva, Plasma [14,18,24,25,26,27,28,29,31]
miR-1275	Tissue [17]	miR-3162	Whole blood [18]
miR-128a	Cell lines [17]	miR-323-5p	Saliva [18]
miR-130b	Tissue, Cell lines [19]	miR-34a	Salivary exosomes [27]
miR-134	Tissue, Plasma [17,24,25]	miR-34b	Tissue, Cell lines [19]
miR-135	Tissue, Cell lines [19]	miR-34c	Tissue, Cell lines [19]
miR-135b-5p	Tissue [17]	miR-3651	Tissue, Whole blood [18,25]
miR-136	Saliva [18]	miR-372	Tissue, Cell lines [24,25]
miR-142	Tissue [17,19]	miR-373	Tissue, Cell lines [24,25]
miR-143	Tissue, Cell lines [19]	miR-412-3p	Saliva [27]
miR-143	Tissue, Cell lines [29]	miR-412-3p	Saliva [18]
miR-144	Tissue [17]	miR-423	Tissue, Cell lines [19]
miR-145	Saliva [27]	miR-423-3p	Tissue, Plasma [18,19]
miR-146a-5p	Tissue, Saliva, Plasma [17,24,26,28,30]	miR-424-5p	Tissue, Cell lines [25]
miR-146b	Tissue [17]	miR-4484	Salivary exosomes [27]
miR-147	Saliva [18]	miR-450a	Tissue, Cell lines [25]
miR-148a	Tissue, Saliva [18,19]	miR-451	Tumor, Saliva, Serum [17]
miR-148b	Cell lines [17]	MiR-4513	Cell lines [25,30]
miR-150-5p	Tissue, Plasma [18,19]	miR-455-5p	Tissue [17,25]
miR-155-5p	Tissue, Cell lines [18,24,26,30]	miR-483	Saliva [18]
miR-15b	Tissue, Cell lines [17]	miR-483-5p	Serum [18]
miR-181	Tissue, Plasma [14,18]	miR-483-5p	Plasma, Serum [17]
miR-181a	Plasma [18]	miR-494	Tissue, Saliva, Whole blood [18,25]
miR-181b	Plasma [18]	miR-497	Tissue [17]
miR-182-5p	Tissue [26]	miR-503	Saliva [18]
MiR-183	Cell lines [30]	miR-5100	Tissue, Serum [18,25]
miR-184	Saliva, Plasma [14,18,27]	miR-512-3p	Saliva [18]
miR-187	Plasma [18]	miR-542	Tissue, Cell lines [19]
miR-18a-5p	Tissue, Cell lines [25]	miR-543	Tissue, Cell lines [25]
miR-191	Whole blood [18]	miR-582-5p	Cell lines [17]
miR-196a-3p	Plasma [32]	MiR-626	Tissue, Cell lines, Serum [18,30]
miR-196a-5p	Saliva, Plasma [18,24,26,31]	miR-632	Saliva [18]
MiR-196b	Tissue, Cell lines, Saliva [24,31]	miR-646	Saliva [18]
miR-196b	Plasma [18]	miR-650	Tissue, Cell lines [25]
miR-200b-3p	Plasma [18]	miR-654	Tissue, Cell lines [25]
miR-21-3p	Tissue [17,25]	miR-668	Saliva [18]
miR-21-5p	Tissue, Cell lines, Saliva, Whole blood Plasma, Serum [17,18,19,25,26,27,28,30,31,32]	MiR-7975	Salivary exosomes [28]
miR-210	Whole blood [18]	miR-877	Saliva [18]
MiR-211	Tissue [14,24,26]	miR-877-5p	Saliva [17]
miR-214	Tissue, Cell lines [17]	miR-92b	Serum [18]
miR-218	Tissue [24,25]	miR-93	Saliva [27]
miR-220a	Saliva [18]	MiR-96-5p	Tissue [26]
miR-221	Tissue, Cell lines [17,19]		

**Table 2 genes-14-01578-t002:** MicroRNA (miRNA) molecules that have been documented to exhibit significantly decreased expression levels in biological samples associated with oral squamous cell carcinomas (OSCCs), such as tumor tissue, OSCC cell lines, saliva, whole blood, plasma, and serum, compared to non-OSCC biological materials.

↓ miRNA	Sample Source	↓ miRNA	Sample Source
let-7a-5p	Tissue, Cell lines, Saliva [17,24,31]	miR-23b-3p	Tissue [17,25,26]
miR-107	Tissue, Cell lines, Saliva [17,27]	miR-26a	Tissue, Cell lines, Saliva [17,24,27]
let-7c	Tissue, Saliva [24,28]	miR-26b	Cell lines [17]
let-7c-5p	Tissue [17]	miR-27a-3p	Tissue, Cell lines [17]
let-7d	Tissue, Cell lines, Saliva, Whole blood, Serum [17,18,24,26]	miR-27b	Tissue, Saliva, Plasma [17,26,27,31]
let-7e	Tissue, Cell lines [24]	miR-299	Tissue, Cell lines [25]
let-7f	Tissue, Cell lines [17,24]	miR-29a-3p	Tissue, Serum [17,18,26,30]
miR-1-3p	Tissue, Cell lines [17,25,26,30]	miR-29b-3p	Tissue, Cell lines [17,30]
miR-100	Tissue, Saliva [17,28]	miR-29c	Tissue, Cell lines [17]
miR-101	Tissue, Cell lines [25,30]	miR-30a-5p	Plasma [18]
miR-106a	Tissue, Cell lines [25]	miR-320	Tissue, Cell lines [25]
miR-107	Tissue, Cell lines [25]	miR-320a	Saliva [18]
miR-10a	Tissue, Cell lines [17]	miR-338-3p	Serum [18]
miR-124-3p	Tissue, Cell lines, Saliva [17,25,31]	miR-340	Tissue [29]
miR-1250	Saliva [17]	miR-34a-5p	Tissue, Saliva [25,27,32]
miR-125a-5p	Tissue, Saliva [14,18,28,31]	miR-375	Tissue, Cell lines, Saliva [19,24,25,26,27,28,30,31]
miR-125b-2-3p	Tissue, Cell lines [17]	miR-376c-3p	Tissue, Cell lines [30]
miR-125b-5p	Tissue, Cell lines [17,25,26]	miR-377	Tissue, Cell lines [25]
miR-125b-5p	Tissue [29]	miR-378	Tissue, Cell lines [30]
miR-126	Tissue, Cell lines [17,25]	miR-4282	Tissue, Cell lines [30]
miR-1271	Tissue, Cell lines [17]	miR-429	Tissue, Cell lines [17,25]
miR-128-3p	Cell lines [17]	miR-433	Tissue, Cell lines [17]
miR-1291	Tissue [17]	mir-4485	Tissue [17]
miR-133a-3p	Tissue, Cell lines [17,25,26,30]	miR-4488	Tissue [17]
miR-133a-5p	Tissue, Cell lines [17]	miR-4492	Tissue [17]
mir-136	Saliva [17,31]	miR-4497	Tissue [17]
miR-137	Tissue, Cell lines [17,24]	miR-4508	Tissue [17]
miR-138-3p	Tissue, Cell lines [17,25]	miR-451	Tissue, Cell lines, Saliva, Serum [17]
miR-138-5p	Tissue, Cell lines [14,17,25,26]	miR-4516	Tissue [17]
miR-139-5p	Tissue, Cell lines, Saliva [17,18,24,25,30,31]	miR-4532	Tissue [17]
miR-141	Tissue [17]	miR-486	Tissue, Cell lines [19,25,30]
miR-142-3p	Saliva [28]	mir-487-3p	Tissue [30]
miR-143	Tissue, Cell lines [17,24,25]	miR-491-5p	Tissue [25,26]
miR-145-5p	Tissue, Cell lines, Saliva [17,18,25,30,31]	miR-494-3p	Cell lines [17]
miR-146a-5p	Tissue, Cell lines, Saliva [25,31]	miR-494-5p	Tissue, Cell lines [17]
miR-147	Saliva [17]	miR-495	Tissue [25,26]
miR-148a	Tissue, Saliva, Plasma [17,26]	miR-499	Tissue [19]
miR-149	Tissue, Cell lines [17,30]	miR-499a	Tissue [17]
miR-150-3p	Tissue, Cell lines [17]	miR-503	Saliva [17]
miR-153-3p	Tissue [26]	miR-504	Tissue [19]
miR-16-5p	Tissue, Cell lines [25,30]	miR-506	Tissue [17]
miR-17-5p	Tissue, Cell lines [25,26,30]	miR-519d	Tissue [26,32]
miR-181a-5p	Tissue, Cell lines, Plasma [25,26]	miR-542-3p	Tissue [17]
miR-184	Tissue, Cell lines [24]	miR-545	Tissue [25,26]
miR-186	Tissue, Cell lines, Whole blood [18,25,30]	miR-585	Cell lines [17]
miR-188	Tissue, Cell lines [25]	miR-6087	Tissue [17]
miR-195	Tissue, Cell lines [25,30]	miR-617	Cell lines [30]
miR-196-5p	Tissue [32]	miR-632	Saliva [17]
miR-196a-5p	Tissue, Cell lines [19]	miR-646	Saliva [17]
miR-198	Cell lines [30]	miR-6510-3p	Tissue [17]
miR-199	Tissue [19]	miR-655	Tissue, Cell lines [25]
miR-199a-5p	Tissue, Cell lines [25,30]	miR-668	Saliva [17]
miR-200a	Saliva, Salivary exosomes [14,18,27,28,31]	miR-675	Tissue, Cell lines [17]
miR-200c	Tissue, Cell lines [25]	miR-7	Saliva [27]
miR-203	Cell lines [25,30]	miR-758	Saliva, Serum [18]
miR-204-5p	Tissue, Cell lines [25]	miR-769-5p	Plasma [18]
miR-205-5p	Tissue, Saliva [25,26,31]	miR-7704	Tissue [17]
miR-214	Tissue [19]	miR-874	Cell lines [17]
miR-216a	Tissue, Cell lines [17]	miR-877-5p	Saliva [17]
miR-218	Tissue, Cell lines, Saliva [17,25,27,29]	miR-9	Tissue, Cell lines, Serum [18,30]
miR-22	Tissue, Cell lines [25]	miR-9	Tissue, Cell lines [25]
miR-22-3p	Tissue, Cell lines [17,30]	miR-92a-3p	Saliva [31]
miR-220a	Saliva [17]	miR-92b	Tissue [19]
miR-221	Tissue, Cell lines [25]	miR-93	Saliva [28]
miR-223	Serum [18]	miR-98	Tissue, Cell lines [25]
miR-23a-3p	Tissue, Cell lines [25]	miR-99a-5p	Tissue, Cell lines, Saliva, Serum [18,24,25,28]

**Table 3 genes-14-01578-t003:** Overview of the results obtained from the miRNA/target prediction analysis. Our analysis focused on specific panels of tumor suppressor genes and oncogenes that have been strongly associated with the development and characteristics of oral oncogenesis. The genes that have been identified as potential targets of each miRNA, following the successful application of our bioinformatic filtering, are indicated in red. Conversely, the genes from each panel that are not targeted are indicated in black. The table also encompasses data pertaining to the expression patterns observed in OSCC for each miRNA molecule, as documented in the latest comprehensive review of the available research. Details are furtherly discussed in the text.

miRNA	Reported Expression in OSCC	Predicted Target OSCC-Associated Tumor Suppressor Genes (5)	Target Score
hsa-miR-155-5p	↑	*TP53*, *CDKN2A*, *FAT1*, *CASP8*, *PTEN*	5/5
hsa-miR-34a-5p	↑ (rarely)	*TP53*, *CDKN2A*, *FAT1*, *CASP8*, *PTEN*	5/5
miRNA	Reported Expression in OSCC	Predicted Target OSCC-associated oncogenes (15)	Target score
hsa-miR-34a-5p	↓ (mostly)	*NOTCH1*, *HRAS*, *PIK3CA*, *EGFR*, *ERBB2*, *FGFR1*, *FGFR2*, *FGFR3*, *FGFR4*, *FGF2*, *ETS1*, *JUN*, *MKI67*, *MYC*, *BCL2*	15/15
hsa-miR-124-3p	↓	*NOTCH1*, *HRAS*, *PIK3CA*, *EGFR*, *ERBB2*, *FGFR1*, *FGFR2*, *FGFR3*, *FGFR4*, *FGF2*, *ETS1*, *JUN*, *MKI67*, *MYC*, *BCL2*	12/15
hsa-miR-1-3p	↓	*NOTCH1*, *HRAS*, *PIK3CA*, *EGFR*, *ERBB2*, *FGFR1*, *FGFR2*, *FGFR3*, *FGFR4*, *FGF2*, *ETS1*, *JUN*, *MKI67*, *MYC*, *BCL2*	10/15
hsa-miR-16-5p	↓	*NOTCH1*, *HRAS*, *PIK3CA*, *EGFR*, *ERBB2*, *FGFR1*, *FGFR2*, *FGFR3*, *FGFR4*, *FGF2*, *ETS1*, *JUN*, *MKI67*, *MYC*, *BCL2*	9/15

## Data Availability

The article includes the original contributions presented in this study. Any further inquiries may be forwarded to the corresponding author.

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
