# Peer review of "Review of Disease-Specific microRNAs by Strategically Bridging Genetics and Epigenetics in Oral Squamous Cell Carcinoma"

_genes, 2023, doi:10.3390/genes14081578_

Round 1

Reviewer 1 Report

In the presented manuscript, the Authors presented key microRNAs in oral squamous cell carcinoma.

1. It is an interesting topic. The manuscript is very informative to follow current updates of miRNAs in oral squamous cell carcinoma.

2. I suggest adding a flow chart to better understand the methodology used by the Authors.

3. The rules for numbering chapters and subchapters are not entirely clear to me. This should be sorted out.

4. It is a pity that the authors removed the numbering lines in the manuscript, which would make it easier to present comments (such lines are available in the Microsoft Word template). The manuscript contains editorial errors e.g. unnecessary double spaces  - ‘…..upregulated in OSCC, respectively.    As a result…..’, ‘miR…. microRNAs (miR NAs)……’ Tables have a different font than the rest of the text. Gene names should be presented in italics.

Author Response

Reviewer 1

Comments and Suggestions for Authors

In the presented manuscript, the Authors presented key microRNAs in oral squamous cell carcinoma.

  1. It is an interesting topic. The manuscript is very informative to follow current updates of

miRNAs in oral squamous cell carcinoma.

We express appreciation to the Reviewer for recognizing our manuscript as a valuable contribution to the existing body of literature.

  1. I suggest adding a flow chart to better understand the methodology used by the Authors.

We would like to thank the Reviewer for the helpful recommendation, which has enhanced the quality of our manuscript. A flow chart has been constructed in order to succinctly outline the sequential procedures of our methodology, thereby improving readers’ comprehension. It is presented as Figure 9.

  1. The rules for numbering chapters and subchapters are not entirely clear to me. This should be sorted out.

We thank the Reviewer for the comment. Indeed, the numbering of certain chapters proved to be more perplexing than conducive to readers’ comprehension. Consequently, we eliminated superfluous numbering and prioritized the use of distinct headings with bold formatting to effectively demarcate sections.

  1. It is a pity that the authors removed the numbering lines in the manuscript, which would make it easier to present comments (such lines are available in the Microsoft Word template). The manuscript contains editorial errors e.g. unnecessary double spaces - ‘…..upregulated in OSCC, respectively. As a result…..’, ‘miR…. microRNAs (miR NAs)……’ Tables have a different font than the rest of the text. Gene names should be presented in italics.

We thank the Reviewer for bringing to our attention the editorial errors that were inadvertently overlooked during the review of the final draft. The aforementioned errors have been rectified and the corresponding corrections have been indicated within the text through precisely placed comments.

Reviewer 2 Report

Dear authors,

It is a very good and interesting review. However, I have some comments.

Introduction

1.     Replace « miR Nas » by miRNAs

Methods

2.     Why did you use only one database?

3.     Could you explain how did you select the 13 articles? Inclusion and exclusion criteria should be presented?

4.     Table 1 Could you check the number of each reference because the number 77 should not be here?

Results

1.     It could be interesting to summarize the results because this section is very long

Discussion

2.     This section is not really necessary. You could summarize the main data to do the conclusion

Author Response

Reviewer 2

Comments and Suggestions for Authors

It is a very good and interesting review. However, I have some comments.

Introduction

  1. Replace « miR Nas » by miRNAs

We thank the Reviewer for finding our review very good and interesting. We also thank the Reviewer for bringing to our attention this particular editorial error that was inadvertently overlooked during the review of the manuscript’s final draft. The aforementioned error has been successfully rectified and the corresponding correction has been indicated within the text through a precisely placed comment.

Methods

  1. Why did you use only one database?

The objective of our study was to develop a focused and strategic review that aims to discern and thoroughly examine the significance of a select group of key miRNAs within the extensive body of literature. If our primary objective had been to comprehensively present all accessible data on miRNA expression in oral cancer, it would have been indeed advisable to consider utilising additional databases. In the current review, we decided to conduct a targeted search using the PubMed database, which is widely acknowledged as the premier source for biomedical literature worldwide, since it provides a publicly accessible search interface for MEDLINE and various other NLM resources. Given the extensive number of findings in this particular research area, we observed that the data retrieved from the searches we conducted using PubMed were abundant and often exhibited significant overlap. Hence, with regards to the study of the literature, which served as a foundation for subsequent analysis, we estimated that the quantity and quality of results obtained from Pubmed, along with the database's renown credibility, has indeed been sufficient for the particular project.

  1. Could you explain how did you select the 13 articles? Inclusion and exclusion criteria should be presented?

We thank the Reviewer for their comment. Those criteria are presented in the “Selection of most significant miRNAs in OSCC” section of the Methodology. More specifically, as stated in text: “The searches yielded 39, 23, and 63 results, respectively, from which we selected 13 articles that were specifically focused on OSCC (excluding precancerous oral pathologies), included more than three implicated miRNA molecules, and clearly stated the expression patterns, as well as the sample sources that corresponded to miRNA quantification results”

  1. Table 1 Could you check the number of each reference because the number 77should not be here?

The review article titled "MicroRNAs as Modulators of Oral Tumorigenesis-A Focused Review" by Rishabh et al. (2021) is referenced as number 77. This article was obtained through our PubMed searches and met all the inclusion criteria outlined in the "Methodology" section. The study conducted by Rishabh et al. (2021) has been referenced multiple times within "Table 1" and "Table 2", since it includes miRNA molecules that are upregulated and downregulated in OSCC-related biological materials. We hereby verify the accuracy of the reference numbers, including reference No. 77 in Table 1.

Results

  1. It could be interesting to summarize the results because this section is very long

We would like to thank the Reviewer for this recommendation. The findings of our analysis are briefly and clearly presented in "Table 3", positioned below the two paragraphs comprising the section titled "Results of bioinformatic analysis of miRNA/target interactions". However, given the intricate study design and methodology employed, which involved a novel approach to discerning crucial miRNAs by identifying their primary targets from customized panels, our aim was to provide a clear and comprehensible body of results in corresponding section. This was performed to facilitate a better understanding of the findings and to ensure the reproducibility of the study for other researchers interested in utilizing this methodology for investigating different pathologies. 

Discussion

  1. This section is not really necessary. You could summarize the main data to do the Conclusion.

It has been estimated that, with the exception of the "Results of bioinformatic analysis of miRNA/target interactions" section, it would be more advantageous to incorporate a "Discussion" section in lieu of "Conclusions." The comprehensive explanation and summary of our results can be found in both the corresponding section and Table 3. A discussion is deemed necessary in light of the objectives of this review, which encompass not only the identification of dysregulated miRNAs in OSCC, but also the proposal of a novel methodology for discerning significant miRNAs within a broad spectrum of molecules that have been associated with a particular condition. This approach holds potential applicability to various pathologies with established genetic backgrounds. The present discussion encompasses an overview of the study's concept, the respective roles of each identified miRNA in OSCC, the significance of this methodology, its broader relevance in the field of cancer research, and ultimately its potential for reproducibility in various diseases, along with future prospects. The examination and analysis of these aspects are of paramount importance for all authors and constitute the main thrust of the study.